# Authors submission guidelines, a survey of pediatric dentistry journals regarding ethical issues

**Tarun Walia**[1]⊙*, **Gauri Kalra**[2]⊙, **Vijay Prakash Mathur**[3]‡, **Jatinder Kaur Dhillon**[4]‡

**1** College of Dentistry, Ajman University, Ajman, United Arab Emirates, **2** Department of Pedodontics & Preventive Dentistry, Sudha Rustagi College of Dental Sciences & Research, Faridabad, Haryana, India, **3** Division of Pedodontics and Preventive Dentistry, Centre for Dental Education and Research, All India Institute of Medical Sciences, New Delhi, India, **4** College of Dental Medicine, Nova Southeastern University, Florida, United States of America

⊙ These authors contributed equally to this work.
‡ These authors also contributed equally to this work.
* ttww@yahoo.com

**Data Availability Statement:** All relevant data are within the manuscript and its Supporting Information files.

**Funding:** The author(s) received no specific funding for this work.

## Abstract

### Objective

To assess the pattern of instructions regarding the ethical requirements given to authors in various Pediatric Dental Journals.

### Material & methods

A cross-sectional survey of 'instructions for authors,' for analysis of guidelines on ethical processes, was done. Instructions to authors in journals of pediatric dentistry across the globe were reviewed for guidelines with regards to fourteen key ethical issues. Descriptive statistics were used, and results were expressed in percentages as well as numbers.

### Results

Of the 18journals of pediatric dentistry, all 14 ethical issues were covered by the instructions to authors in only three journals with only 50% of these providing clarity about authorship using ICMJE guidelines. Furthermore, COI declaration was found to be present as mandatory in about 44% of the journals. 38.9% of the sampled journals mentioned guidelines on research misconduct, publication issues such as plagiarism, overlapping/fragmented publications, and availability of raw research data from authors. Guidelines on handling of complaints about editorial team was provided to authors by slightly over 33% of the selected pediatric dentistry titles while handling of complaints about authors and reviewers were mentioned in 16.7%and 55.6% of the journals respectively.

### Conclusion

A significant proportion of Journals of Pediatric Dentistry did not provide adequate instructions to authors regarding ethical issues.

**Competing interests:** The authors have declared that no competing interests exist.

## Introduction

Ethics in biomedical research and publications have always been a topic of deliberation. Ethical process of data collection and validation of clinical research carries great significance in scientific endeavors. Any inadequacy in following ethical principles during the conduct of a research and its reporting may affect the field of research and practice immensely. In order to formulate guidelines for transparent and ethical research conduct, the Declaration of Helsinki was formulated by the World Medical Association whichincluded32 principles covering informed consent, data confidentiality, vulnerable population and protocol requirement for conducting research and ethics committee approval [1]. These guidelines are revised periodically, and the most recent revisions took place in year 2013 suggesting the addition of five more principles which included use of placebo, post-trial provisions, trial registry, publication & dissemination of results and application of unproven interventions in clinical practice [2]. In association with World Health Organization, the Council for International Organizations of Medical Sciences (CIOMS) in 2017 redefined "International Ethical Guidelines for Biomedical Research Involving Human Subjects" [3]. Recently in 2021, the Committee for the Purpose of Control and Supervision of Experiments on Animals (CPCSEA) also revised its extent of guidelines for carrying out experiments on animals [4]. These recommendations are developed to review best practice and ethical standards in the conduct and dissemination of research.

Reporting of research work in the form of publication in a peer reviewed journal is imperative for propagation of science as well as recognizing timely developments. International Council for Medical Journal Editors (ICMJE) too has updated the recommendations for the Conduct, Reporting, Editing, and Publication of Scholarly Work in Medical Journals in the year 2019 [5]. These guidelines were developed to encourage researchers to publish scientific research with accepted and approved ethical criteria. The new changes also, provided useful insights to the editors with biomedical editing and publication process [6]. Some of the medical journals remain up to date about the amendments and accordingly revise the instructions to authors but few of them do not update the guidelines. The latter remain implicit on reporting policies about ethical principles and only mention that they follow 'ICMJE Guidelines'. Mentioning of specific submission guidelines for authors on the journal websites and implied statements particularly on the ethical principles does not astound upon the authors to follow the established protocols.

The present group of authors in 2012 published a research that evaluated and compared author instructions in Indian and the British journals in the field of pediatric dentistry [6]. The study brought to light the lacunae in manuscript submission guidelines for the authors/researchers in the selected journals. However, after the recent amendments in ICMJE guidelines, in 2017, which recommend an active implementation of reporting guidelines in scientific journals, it becomes imperative to assess the inclusion of aforesaid reporting guidelines in author submission instructions of dental journals.

The aim of the present study was to assess the pattern of submission guidelines regarding the ethical issues given to authors in various pediatric dental journals.

## Material & methods

### Inclusion & exclusion criteria

Following databases were included to search all available titles in the field of pediatric dentistry:

1. PubMed/National Library of Medicine (NLM)

2. Web of Science

3. Scopus/Scimago

4. Index Copernicus

5. COSMOS

6. Directory of Open Access Journals (DOAJ)

7. EMBASE

Pediatric dental journals cited in at least anyone of the above-mentioned databases were included. However, the journals that was discontinued or with irregular publishing was excluded. The screening of the journal names was performed in English language. Journals in non-english publishing language were also included in the survey if the instructions to authors were available in English. During journal search, textbooks, Practical reviews, and newsletters were also excluded.

## Search strategy

A search strategy was planned in January 2021to find out the list of published journals in the specialty of pediatric dentistry across the world. Initially, the broad-based search was implemented individually with keywords: "Paediatric Dentistry", "Pediatric Dentistry", "Pedodontics" and "Dentistry for Children". The country specific pediatric dentistry association websites were accessed through National Member Societies (NMS) of International Association of Pediatric Dentistry (IAPD) and searched for their official publication. Two authors (TW and GK) performed this search independently according to this predefined strategy. Then they shared their search lists and independently removed duplicates on their own and shared their search lists with the third author (JK) for cross checking. Only one difference was observed between both the searches which were resolved with dialogue between the three authors (JK, TW and GK) to have a finalized list. The authors could not be blinded at any stage of survey due to the design of study.

## Retrieval of instructions to authors

After freezing the list of journals, instructions to authors were downloaded into computer by the two authors (GK and TW). The downloaded files were then shared for further analysis of instructions to authors.

The checklist for marking compliance was mainly used from the previous article of same authors [6] However, after reviewing the new guidelines, it was modified to evaluate the following fourteen parameters in each of the selected journal [5]:

1. For authorship- guidelines of International Committee of Medical Journal Editors (ICMJE).

2. Conduct of study—Committee on Publication Ethics (COPE) or mention of reporting criteria for specific type of studied.

3. Approval from an institutional/independent ethics committee mandatory.

4. Details about requirement for obtaining informed consent and maintenance of confidentiality.

5. Mention about animal welfare.

6. Mandatory declaration of conflicts of interest (COI) by authors,

7. Mention about journal policies on publication issues (redundant, fragmented or overlapping publications, and plagiarism).

8. Mention about journal policies on any other research misconduct.

9. Mention about journal guidelines on handling of complaints about authors.

10. Mention about journal guidelines on handling of complaints about reviewers.

11. Mention about journal guidelines on handling of complaints about editorial team.

12. Mention about journal guidelines on handling of Authorship disputes.

13. Availability of raw data in case of trials which may require cross evaluation.

14. Copyright related issues.

### Data entry about ethical issues

After carefully reading the instructions to authors for each journal, GK and TW entered the score into the MS Excel sheet based on the above-mentioned parameters. The authors also recorded additional findings and mentioned about adherence to some specific guidelines if any. Both authors also entered the impact factor (JCR 2020) for the included journals (for which it is available) (Table 1).

**Table 1. List of included pediatric dentistry journals.**

| S. No. | Journal Name | Affiliation to Professional Association/ society | Number of ethical principles mentioned in the submission guidelines to authors. | Impact factor |
|---|---|---|---|---|
| 1. | European Archives of Pediatric dentistry | European Academy of Pediatric Dentistry | 13 | - |
| 2. | European journal of pediatric dentistry | Italian Society of Pediatric Dentistry | 3 | 2.231 |
| 3. | International Journal of Clinical Pediatric Dentistry | - | 5 | - |
| 4. | International Journal of Pediatric Dentistry | International Association of Pediatric Dentistry | 11 | 3.455 |
| 5. | International Journal of Pedodontic Rehabilitation | - | 7 | - |
| 6. | Interventions in Pediatric Dentistry Open Access Journal | - | 2 | - |
| 7. | Journal of Clinical Pediatric Dentistry | - | 3 | 1.065 |
| 8. | Journal of Dentistry for Children | American Academy of Pediatric Dentistry | 14 | - |
| 9. | Journal of Indian Society of Pedodontics and Preventive Dentistry | Indian Society of Pedodontics and Preventive Dentistry | 6 | - |
| 10. | Journal of South Asian Association of Pediatric Dentistry | South Asian Association of Pediatric Dentistry | 8 | - |
| 11. | OdontologiaPediaťrica | - | 1 | - |
| 12. | Paidodontía | - | 2 | - |
| 13. | Pediatric Dental Journal | Japanese Society of Pediatric Dentistry and The Pediatric Dentistry Association of Asia | 12 | - |
| 14. | Pediatric Dentistry | American Academy of Pediatric Dentistry | 14 | 1.874 |
| 15. | PesquisaBrasileiraemOdontopediatria e ClínicaIntegrada (Brazilian Research in Pediatric Dentistry) | - | 3 | - |
| 16. | RevistaLatinoamericana de ortodoncia y odontopediatría. | - | 14 | - |
| 17. | Shōnishikagakuzasshi. The Japanese journal of pedodontics | Japanese Society of Pediatric Dentistry and The Pediatric Dentistry Association of Asia | 12 | - |
| 18. | TaehanSoaChʻikwaHakhoe chi = Journal of the Korean Academy of Pedodontics. | Korean Academy of Pedodontics. | 4 | - |

### Cross check of data entered

The downloaded instructions to authors and scores entered against each parameter were then shared with JK and VM for cross check and verification. There were few opinion differences and were resolved by email exchanges and discussion. If policies/guidelines of a journal were not clearly mentioned, or the language was confusing for a particular parameter then it was considered as partial compliant.

### Statistical analysis

The data was entered in Microsoft excel 2016 and analyzed descriptively using SPSS software (V 22.0).

## Results

Initial screening of pediatric dentistry journals retrieved fifty-four national and international journals titles from the seven selected search databases. After removal of duplicate and discontinued journals, eighteen journals were included in the final list for this bibliometric study (Table 1). The detailed steps of literature search and results has been described in Fig 1. Out of 18 journals, instructions to authors were accessed from independent journal websites for 12 journals while instructions for two journals were available at the publisher's website. Author guidelines for the remaining 4journals were retrieved from their society website. Only three journals, namely Journal of Dentistry for Children, Pediatric Dentistry and RevistaLatinoamericana de ortodoncia y odontopediatría covered all fourteen parameters related to ethical issues in the submission guidelines for authors. Out of 18, only 4 journals had an impact factor (Table 1). All these four journals had an impact factor of more than 1 (1.065 to 3.455).

The various parameters included in the study have been defined descriptively in Fig 2. Two-thirds (14) of the journal's mentioned reporting guidelines/criteria for conduct of the various study types, ethical approval from IRB/Ethics Review Committees and copyright related issues. About 16 Pediatric dental journals mentioned the clause for animal welfare and had an online form declaring the same available for download. Ethical constraints such as guidelines on authorship criteria were mentioned in the instructions provided by 9of the journals. Handling author disputes and raising complaints regarding peer-review process, necessity to obtain an informed consent and maintaining participant confidentiality were mentioned in 10of the included journals. Furthermore, COI declaration was found to be present as mandatory in about 8of the journals. Seven of the sampled journals mentioned guidelines on research misconduct, publication issues such as plagiarism, overlapping/fragmented publications, and availability of raw research data from authors. Guidelines on handling complaints about editorial team were provided to authors by only 6of the selected Pediatric dentistry titles. While handling complaints about authors and reviewers were mentioned in only 3 and 10 of the journals respectively.

## Discussion

ICMJE has laid transparent ethical recommendations for scientific publications; however, the self-monitoring by the countries and scientific organizations with respect to these guidelines has not been well documented. Such reporting guidelines serve as important tools of reference for systematic ethical report writing. Therefore, these guidelines in form of instructions to authors must be present either in form of checklists, flowcharts, or simple texts on accessible portals of all scientific journals. Inconsistency in the submission guidelines and instructions to authors particularly in context of recent amendments [2–4] amongst various dental and

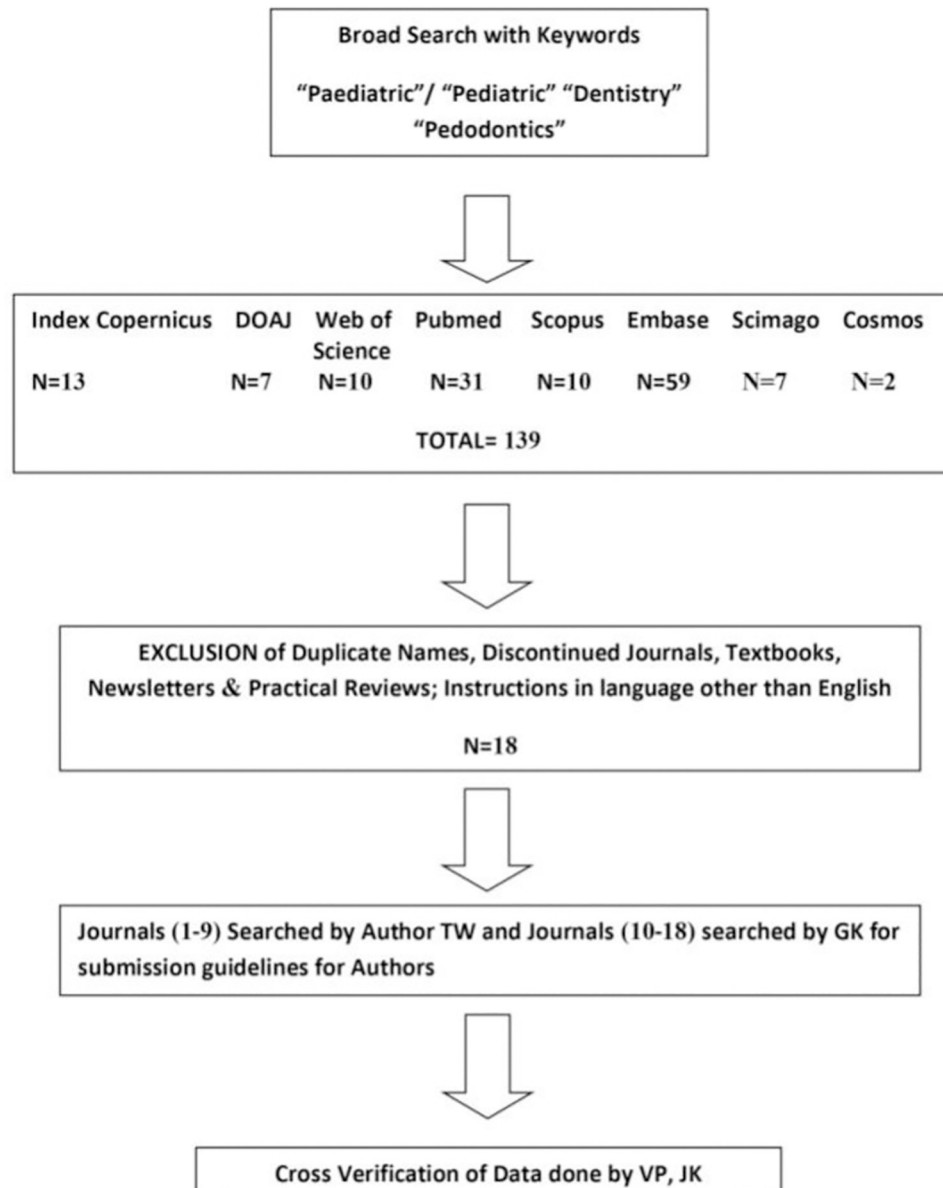

**Fig 1. Literature search methodology and results.**

medical journals led us to test whether inclusion of ethical issues is being followed by all published scientific journals in the subject of pediatric dentistry. In the previous study, the number of journals covered were limited to only two countries [6] (India and Britain) whereas the present study was planned to evaluate pediatric dentistry journals globally. The findings from the current study suggests that ethical principles as per the recent amendments by ICMJE and COPE are still not clearly defined in the pediatric dental journals that are presently in circulation.

Authorship policies have been evidently described by ICMJE to promote integrity and accountability in research. In the present study, 50% (9) of the journals didn't mention authorship or contributor ship criteria in their author instructions. Our findings are concurrent with the results from a study conducted by Resnik DB and colleagues who found 63% of 600 journals sampled from Journal Citation Reports Database, had listed authorship policies. They also

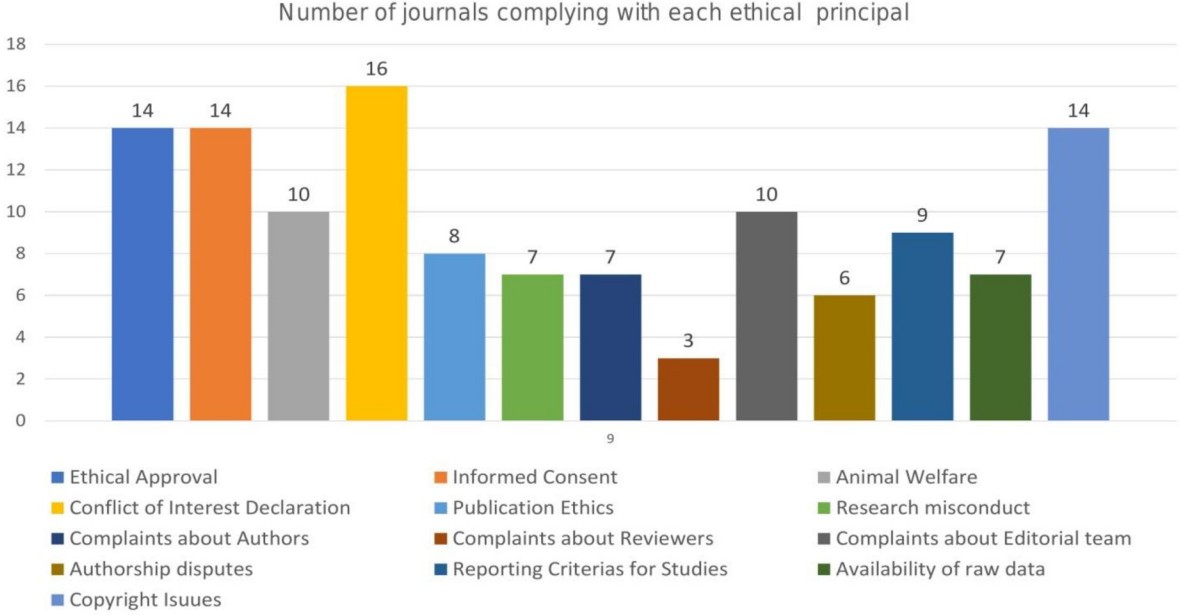

**Fig 2. Number of journals complying with each ethical principal.**

concluded that journals from the biomedical sciences and social sciences/humanities had more chances of having an authorship policy compared to the physical sciences, engineering, or mathematical sciences journals [7]. It becomes imperative for the journal editors to endorse multiple reporting guidelines to aid researchers in preparing specific type of research and improve research documentation in form of flowcharts as well as checklists. In the present study, only 3 of the selected eighteen pediatric dentistry journals were found totally compliant in reporting various types of manuscripts reporting guidelines. Previous published literature too has described suboptimal reporting by medical journals on various guidelines such as STROBE PRISMA, CONSORT etc. This signifies low endorsement of guidelines on the conduct of study and its types [8].

Furthermore, Article 23 of DoH (Declaration of Helsinki) has clearly defined the important role and function of research ethics committee or an independent institutional review board. It informs review of the scientific protocols before beginning any research involving human subjects. Also, Article 25 to Article 32 of DoH significantly explains the requirement of taking a patient informed consent before making the subject a part of a research [2]. In the current study, 14 out of 18 (78%) journals mentioned instructions about taking approval from Institute Ethics Committee or a Review Board before commencement of the study. These journals also mentioned to provide ethical reference number of approvals from respective ethics committee or requested a copy of the same. Similarly, 78% (14) of the included journals have explicitly mentioned in submission guidelines for authors to state an informed consent from participants. About 56% of the pediatric dentistry journals showed compliance with inclusion of animal welfare. Navaneetha, reported that only 45.2% (57/126) indexed international journals provided instructions to authors reporting ethical approval, 30% insisted upon taking an informed consent and only 26% mentioned about the animal welfare [9]. On the contrary, in the previous study conducted by the present authors, protection of human and animal welfare, informed consent and maintaining the confidentiality of the study subjects was reported higher upto80% in Indian dental journals and 70.3% in British dental journals [6].

As the health care professionals indulge themselves in unending research, in a race to publish maximum research articles, incidence of inappropriate research practices has been reported to be on increase [10,11]. It becomes essential to keep a check on misconduct in research or during its publication. ICMJE have published the best guidelines for refraining oneself from any misconduct for authors, reviewers, and journal editors. Areas of misconduct include plagiarism, redundant publication, data falsification/fabrication, authorship malpractices such as ghost authorships, gift authorships; non-disclosure of conflicts of interest by the authors, delayed publication, and salami publication [12]. Amongst all, Conflict of interest (COI) has been a critical parameter whose non-disclosure may lead to breach of scientific sanctity of a publication. In the present study COI was found to be mentioned in majority of instructions to authors section in eighteen selected pediatric dental journals. This was concurrent with the findings of a previous research that compared Indian and British dental journals [6]. Failure to disclose COI may lead to publication bias and affect the impact factor of reputed journals. However, Jiayi Zhu &Ji Sun (2019) reported that Chinese medical journals are not adequately emphasizing on declaration of Conflict of Interest (COI) [13]. This may be due to overemphasis on more important aspects of scientific learning such as plagiarism, multiple submissions, redundant publications etc. rather than disclosure policies. Sahni et al (2018) mentioned that almost all scientific journals ask for author's copyright transfer to the publisher [14]. However, in the current study, almost two-thirds of pediatric dental journals surveyed have been found to have copyright transfer policy which is likely to reduce any future author-publisher issues. Availability of research raw data must be accessible to the readers to enhance the transparency of research; therefore, many journals demand the researchers to make their raw data accessible whenever required [15]. Nonetheless, in the current study, only one-third of pediatric dentistry journals asked for keeping raw research data in their instructions. The ICJME Uniform Requirements for Manuscripts Submitted to Biomedical Journals has mentioned the availability of raw clinical data taking care of protection of patient privacy and maintaining confidentiality [16]. The transparent process of resolution of disputes and addressing complaint in the journal office are an important requirement in modern day science. It is expected that journals should declare their policies in a clear and transparent manner, however in the present study only a little over half of the journals (55%) have evidently mentioned it in their author guidelines.

The current study suggests that the essential constraints of ethical research conduction and publication are not mentioned completely within all the indexed/non-indexed pediatric dental journals as per the recent guidelines issued by ICMJE thereby highlighting the lacunae in transparent, precise, and worthy publications in scientific research. One of the limitations of this study was that it only focused on the presence of ethical constraints necessary for scientific research publication and not reflecting the overall editorial processes of the respective journals. In a wishful note, the international conglomeration of professional societies of pediatric dentistry should make their own elaborate guidelines for their journals covering ethical aspects and publication ethics with transparency and medical principles at the base.

## Conclusions

- Ethical issues such as ethical approval, informed consent, patient confidentiality; research misconduct is being covered in majority of pediatric dental journals in the current study.

- Areas of data availability, author disputes, and complaint against authors, reviewers and editors are not being emphasized adequately in submission guidelines for authors in various pediatric dentistry publications.

- Ethical issues regarding institutional review board, conduct of research as per guidelines, animal welfare and copyright issues have been sufficiently covered in these selected titles.

- Although, when detailed instructions were not available for any of the ethical construct, the external link to international/national bodies governing the ethical issues were mentioned.

- There is a need that journals must carefully cover all aspects on ethical conduct and research reporting in submission guidelines for authors.

## Supporting information

**S1 Appendix.**
(DOCX)

## Acknowledgments

Deanship of Graduate Studies and Research, Ajman University, United Arab Emirates supported the open access publication fees.

## Author Contributions

**Conceptualization:** Tarun Walia, Gauri Kalra, Vijay Prakash Mathur, Jatinder Kaur Dhillon.

**Data curation:** Tarun Walia, Gauri Kalra, Vijay Prakash Mathur, Jatinder Kaur Dhillon.

**Formal analysis:** Tarun Walia, Gauri Kalra, Vijay Prakash Mathur, Jatinder Kaur Dhillon.

**Investigation:** Tarun Walia, Jatinder Kaur Dhillon.

**Methodology:** Tarun Walia, Gauri Kalra, Vijay Prakash Mathur.

**Resources:** Tarun Walia, Gauri Kalra, Jatinder Kaur Dhillon.

**Validation:** Tarun Walia, Gauri Kalra, Vijay Prakash Mathur.

**Writing – original draft:** Tarun Walia, Gauri Kalra, Vijay Prakash Mathur, Jatinder Kaur Dhillon.

**Writing – review & editing:** Tarun Walia, Gauri Kalra, Vijay Prakash Mathur, Jatinder Kaur Dhillon.

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
