## [Decision Letter · Decision Letter 0]

15 Jul 2021

PONE-D-21-18572

Survey of Submission Guidelines for Authors on Ethical Issues in Pediatric Dentistry Journals

PLOS ONE

Dear Dr. Walia,

Thank you for submitting your manuscript to PLOS ONE. After careful consideration, we feel that it has merit but does not fully meet PLOS ONE’s publication criteria as it currently stands. Therefore, we invite you to submit a revised version of the manuscript that addresses the points raised during the review process.

We look forward to receiving your revised manuscript.

Kind regards,

Despina Koletsi, Dipl.D.S, MSc, Dr. med. dent, MSc, DLSHTM, PGCHEd

Academic Editor

PLOS ONE

Reviewers' comments:

Reviewer's Responses to Questions

**Comments to the Author**

1. Is the manuscript technically sound, and do the data support the conclusions?

Reviewer #1: Partly

Reviewer #2: Yes

2. Has the statistical analysis been performed appropriately and rigorously? 

Reviewer #1: No

Reviewer #2: Yes

3. Have the authors made all data underlying the findings in their manuscript fully available?

Reviewer #1: No

Reviewer #2: Yes

4. Is the manuscript presented in an intelligible fashion and written in standard English?

Reviewer #1: Yes

Reviewer #2: Yes

5. Review Comments to the Author

Reviewer #1: The authors have to decide which terminology to use, and apply it through out the text, is it ethical requirements, guidelines or principles?

Title to be revised: Authors ethical issues submission guidelines, a survey of Pediatric Dentistry journals.

Aims: Missing from document, currently only in the abstract, thus to be added in the document.

Methods and Materials:

How was the checklist compiled, was it based on the ICJME document? Which references were used for? To add at page 6 line 111.

Was the checklist/ survey been validated priorly of doing the survey? What was the % of disagreement between the two evaluators? Was disagreement found mostly on one specific or several items of the check list?

Results

Raw results per Journal and per principle to be included either as Descriptive Table or as appendix.

Table 1, to add on an extra column the Impact Factor of the Journals, for those that it exists.

Table 2, could be graphically presented in a horizontal bar histogram for each item of the checklist, to make results easier to apprehend.

Additional analysis:

1) The check list could be grouped in 3 categories, by weight of contribution and then the results presented accordingly by histogram or pie chart.. Check list categories to be considered are: A=1 through 6, B=7-12, C=13,14.

This type of analysis will provide some weight on the ethical principles missing per journal.

2) The association of the Impact Factor IF on the ethical principles , to be investigated.

Conclusions: To be revised as an answer to the aim.

Reviewer #2: The present article deals with a very important and interesting issue and highlights the fact that journals must carefully cover all aspects on ethical conduct and research reporting in submission guidelines for authors. However, I believe that with some changes, it can be better presented. My detailed comments can be found in the attached file.

6. PLOS authors have the option to publish the peer review history of their article (what does this mean?). If published, this will include your full peer review and any attached files.

Reviewer #1: No

Reviewer #2: No

---

## [Author Response · Author response to Decision Letter 0]

25 Aug 2021

Reviewer #1: 

Comment: The authors have to decide which terminology to use, and apply it throughout the text, is it ethical requirements, guidelines or principles? – submission guidelines as per Ethical Principals

Response: Author submission guidelines - used throughout and has been modified throughout the manuscript

Comment: Title to be revised: Authors submission guidelines, a survey of Pediatric Dentistry journals regarding ethical issues 

Response: Title modified (Lines 1 and 2)

Comment: Aims: Missing from document, currently only in the abstract, thus to be added in the document. 

Response: Aims & Objectives added (Lines 73-74)

Comment: Methods and Materials: How was the checklist compiled, was it based on the ICJME document? Which references were used for? To add at page 6 line 111. Was the checklist/ survey been validated priorly of doing the survey? 

Response: The checklist for marking compliance was mainly used from the previous article of same authors-Mathur VP, Dhillon JK, Kalra G, Sharma A, Mathur R. Survey of instructions to authors in Indian and British Dental Journals with respect to ethical guidelines. J Indian Soc PedodPrev Dent. 2013;31(2):107-112. Modifications were done in context with recent amendments in ICMJE guidelines- Recommendations for the Conduct, Reporting, Editing, and Publication of Scholarly Work in Medical Journals. http://www.icmje.org/icmje-recommendations.pdf. Reference added - line 105-107 and Lines 108-110.

Comment: What was the % of disagreement between the two evaluators? Was disagreement found mostly on one specific or several items of the checklist? 

Response: Disagreement between evaluators were not tested as the findings were cross verified by a group of other two authors 

Comment: Results - Raw results per Journal and per principle to be included either as Descriptive Table or as appendix. 

Response: Raw results/ data entry may be shared only for perusal of editors and reviewers only

Comment: Table 1, to add on an extra column the Impact Factor of the Journals, for those that it exists. 

Response: Added

Comment: Table 2, could be graphically presented in a horizontal bar/ histogram for each item of the checklist, to make results easier to apprehend. 

Response: Figure 2 added

Comment: Additional analysis: 1) The checklist could be grouped in 3 categories, by weight of contribution and then the results presented accordingly by histogram or pie chart. Checklist categories to be considered are A=1 through 6, B=7-12, C=13,14. This type of analysis will provide some weight on the ethical principles missing per journal.

Response: The bar graph has been described which is self explanatory for individual ethical principles in Figure 2

Comment: 2) The association of the Impact Factor IF on the ethical principles, to be investigated.

Response: Calculating association is not possible as not all journals have an impact factor, however, it has been mentioned in table 1.

Reviewer #2: 

Comments: SECTIONS - According to the submission guidelines of the journal, the major sections of the article should be as follows: ABSTRACT, INTRODUCTION, MATERIALS AND METHODS, RESULTS, DISCUSSION, CONCLUSIONS. Τhe “MATERIALS AND METHODS” section has been omitted. Please add this major section and use sub-sections for search strategy e.t.c. Please use the same format for the sections of the same level, as presented in the sample manuscript body included in the Journal Submission Guidelines. 

Response: Appropriate sections have been mentioned with subsections in the Materials and Methods section. Abstract- line 17, Introduction- line 36, Aims & objectives - line 73, Materials and methods- line 75, Results- line 144, Discussion- line 183 and Conclusions- line 268.

Comments: ABSTRACT - The “Background” is not necessary according to the Journal Guidelines in the Abstract Session and it can be omitted. The Objective of the study must be clearly reported in a separate subsection. The 3 subsections of Settings and Design, Materials and Methods and Statistical Analysis can be included all in a single subsection “Materials and Methods. 

Response: Headings in the abstract modified. Changes done lines 17 to 34

Comments: INTRODUCTION - In this section, the significance of the present study and the reason why the problem addressed is important are well presented. However, no reference is made to the existing literature or whether similar studies have been carried out. Ιt is not clear what is new that this study offers in the existing literature. Nor is the purpose of the study clearly stated. This information could be included in two additional separate paragraphs.

Response: The reason to conduct this study has been highlighted in the last para of introduction. Reference study mentioned. Changes from lines 66 to 74. 

Comments: MATERIALS AND METHODS

(i) The inclusion and exclusion criteria of the Journals could be presented more clearly. 

Response: Mentioned. Lines 76-90.

(ii) Was the authors' blindness to the Journal somehow ensured? If not, it must be mentioned. 

Response: Authors could not be blinded due to the design of study. Lines 102-103

(iii) The statistical analysis performed should be reported at the end of this section in a separate paragraph.

Response: Mentioned. Lines 142-143

Comments: RESULTS – (i) Concerning the search results, a flow chart would provide more information and be more explanatory. 

Response: Flowchart for search results added. Figure 1.

(ii) The results should be presented by absolute number too and not only by percentages.

Response: Results presented as absolute numbers in text. Lines 147-167

(iii) Table 1 or Table 2 could be presented as a bar graph instead of a table for better representation of the results graphically. 

Response: Bar graph for table 2 added. Figure 2.

(iv) In the text, there is no need to mention the percentages of compliant Journals for all 14 principles, as these are shown in the table. However, the emphasis should be given mainly to the most or least frequent ethical principles not mentioned in the Pediatric Dentistry Journals. 

Response: Results presented as absolute numbers in text. Lines 157-171.

Comments: DISCUSSION – (i) This section is well written, and all aspects are analyzed correctly. In the second paragraph, reference citation “7” is repeated and a citation is missing at the end of this paragraph.

Response: Reference 7 deleted; reference 8 added at end of this paragraph. Line 211

Comments: CONCLUSIONS – (i) Line 259 Please start with a new sentence “…Ethical issues…” (ii) It would be preferable and more readable if you could present the conclusions using bullets instead of a single paragraph. 

Response: Started with desired line. Conclusion presented in bullets. Lines 269-279. 

Comments: TABLES - Please use the same format for table titles, as follows: “Table 1. This is the Table 1 Title.”

Response: Format corrected. Line 176

Comments: REFERENCES – (i) Please cite references in brackets (for example, “[1]” or “[2-5]” or “[3,7,9]”). 

Response: References cited in brackets. Cited throughout the manuscript

Comments: (ii) There is inconsistency in the references. Firstly, references with more than six authors should list the first six author names, followed by “et al.” Moreover, all references should follow exactly the same style.

Response: Referencing of authors changed as per instructions. References 8, 15, 16

---

## [Decision Letter · Decision Letter 1]

1 Dec 2021

PONE-D-21-18572R1Authors submission guidelines, a survey of Pediatric Dentistry journals regarding ethical issues.PLOS ONE

Dear Dr. Walia,

Thank you for submitting your manuscript to PLOS ONE. After careful consideration, we feel that it has merit but does not fully meet PLOS ONE’s publication criteria as it currently stands. Therefore, we invite you to submit a revised version of the manuscript that addresses the points raised during the review process.

We look forward to receiving your revised manuscript.

Kind regards,

Despina Koletsi, Dipl.D.S, MSc, Dr. med. dent, MSc, DLSHTM, PGCHEd

Academic Editor

PLOS ONE

Journal Requirements:

Reviewers' comments:

Reviewer's Responses to Questions

**Comments to the Author**

1. If the authors have adequately addressed your comments raised in a previous round of review and you feel that this manuscript is now acceptable for publication, you may indicate that here to bypass the “Comments to the Author” section, enter your conflict of interest statement in the “Confidential to Editor” section, and submit your "Accept" recommendation.

Reviewer #1: (No Response)

Reviewer #2: All comments have been addressed

2. Is the manuscript technically sound, and do the data support the conclusions?

Reviewer #1: Yes

Reviewer #2: Yes

3. Has the statistical analysis been performed appropriately and rigorously? 

Reviewer #1: Yes

Reviewer #2: Yes

4. Have the authors made all data underlying the findings in their manuscript fully available?

Reviewer #1: No

Reviewer #2: Yes

5. Is the manuscript presented in an intelligible fashion and written in standard English?

Reviewer #1: Yes

Reviewer #2: Yes

6. Review Comments to the Author

Reviewer #1: Please state the reasons the raw data of your work cannot be made public as an Appendix to this publication

Reviewer #2: Dear authors,

You have adequately addressed generally all the comments raised in the previous round or review process. In the attached file you can see some additional comments.

7. PLOS authors have the option to publish the peer review history of their article (what does this mean?). If published, this will include your full peer review and any attached files.

Reviewer #1: **Yes: **Katerina Kavvadia

Reviewer #2: No

---

## [Author Response · Author response to Decision Letter 1]

4 Dec 2021

REVIEWER´S COMMENTS#1: Please state the reasons the raw data of your work cannot be made public as an Appendix to this publication 

AUTHOR´S REPLY: Raw data has been uploaded as a separate file with main manuscript. 

Changes done on page/line number: A separate document named as Appendix attached.

REVIEWER´S COMMENTS# 2: In the abstract section, the “background” was correctly deleted and it is now well structured. However, I would recommend to replace “Aim and Objectives” with just “Objective”. 

AUTHOR´S REPLY: Aim and Objectives” changed to “Objective”. 

Changes done on page/line number: Line 17

REVIEWER COMMENTS # 2: The significance of the study and the reason of its conduction are now well highlighted in the Introduction section. However, in the Aim and Objective section, it would be preferable if there was no title and just the beginning of the sentence was: “The aim of the present study was to assess the pattern of submission guidelines regarding the ethical issues given to authors in various Pediatric Dental Journals.“ 

AUTHOR´S REPLY: Changes done. 

Changes done on page/line number: Line 73-74

REVIEWER´S COMMENTS: The subsections of the MATERIALS AND METHODS section should follow the same format. All for example bold and italics. 

AUTHOR´S REPLY: Changes done. Sub headings changed in bold and italic. 

Changes done on page/line number: Line 76, 91, 104, 129, 135, 141.

REVIEWER´S COMMENTS: Please rephrase lines 85-86: “However, the journals which were discontinued or with irregular publishing were excluded.” 

AUTHOR´S REPLY: Changes done. Language changed. 

Changes done on page/line number: Line 86.

REVIEWER´S COMMENTS: In line 93, the keyword “Pediatric Dentistry” is mentioned twice. 

AUTHOR´S REPLY: Two different key words used are- Paediatric Dentistry and Pediatric Dentistry due to differences in British and American English; spellings corrected. 

Changes done on page/line number: Line 93.

REVIEWER´S COMMENTS: In the figure and table legends please delete the phrase: “This is the figure 1…” or “This is the Table 1…”. For Figure 1, you can just write: “Figure 1: Literature Search Methodology and Results.” 

AUTHOR´S REPLY: Legends for table and figures corrected. 

Changes done on page/line number: Line 172, 176, 181.

REVIEWER´S COMMENTS: In conclusions, in the second bullet please split the sentence in order to be clearer. 

AUTHOR´S REPLY: The second bullet has been split into different sentences as second and third bullet for clarity. 

Changes done on page/line number: Line 271-276.

REVIEWER´S COMMENTS: In conclusions, the following sentence: “There is a need that journals must carefully cover all aspects on ethical conduct and research reporting in submission guidelines for authors.” could be placed in a separate bullet. 

AUTHOR´S REPLY: Bullet separated for “There is a need that journals must carefully cover all aspects on ethical conduct and research reporting in submission guidelines for authors.” in conclusion. 

Changes done on page/line number: Line 280- 281.

REVIEWER´S COMMENTS: Please check again the references so that all follow the same format. Inconsistencies exist again, so make sure that the dates, the authors etc. are written in the same way. 

AUTHOR´S REPLY: References checked and modified. 

Changes done on page/line number: Format of References number 1,2,6,7,8,9, 13,14, 16 modified.

---

## [Editor Report · Decision Letter 2]

9 Dec 2021

PONE-D-21-18572R2Authors submission guidelines, a survey of Pediatric Dentistry journals regarding ethical issues.PLOS ONE

Dear Dr. Walia,

Thank you for submitting your manuscript to PLOS ONE. After careful consideration, we feel that it has merit but does not fully meet PLOS ONE’s publication criteria as it currently stands. Therefore, we invite you to submit a revised version of the manuscript that addresses the points raised during the review process.

We look forward to receiving your revised manuscript.

Kind regards,

Despina Koletsi, Dipl.D.S, MSc, Dr. med. dent, MSc, DLSHTM, PGCHEd

Academic Editor

PLOS ONE

Journal Requirements:

Additional Editor Comments (if provided):

Dear authors,

After close evaluation of the manuscript and the uploaded raw data, I have noticed that the Impact Factor presented for the journals is not correct (both in raw data information and Table 1). It appears that you have mixed- up Impact scores of some journals with impact factors.

For example, European Archives of Paediatric Dentistry has no Impact Factor currently (Clarivate Analytics), but reports an Impact score of 2.16. this is different. pls correct throughout and re- examine closely for all journals. As in others, for example Intern J Paediatr Dent, the correct IF is given. Also, for others there is inconsistency on what it is reported in the Table and Appendix file with raw data.

Please use the Clarivate Analytics of the Web of Science Journal Citation Reports 2020, consistently.

Thank you.
---

## [Author Response · Author response to Decision Letter 2]

10 Dec 2021

Additional Editor Comments:

After close evaluation of the manuscript and the uploaded raw data, I have noticed that the Impact Factor presented for the journals is not correct (both in raw data information and Table 1). It appears that you have mixed- up Impact scores of some journals with impact factors. For example, European Archives of Paediatric Dentistry has no Impact Factor currently (Clarivate Analytics), but reports an Impact score of 2.16. this is different. pls correct throughout and re- examine closely for all journals. As in others, for example Intern J Paediatr Dent, the correct IF is given. Also, for others there is inconsistency on what it is reported in the Table and Appendix file with raw data. Please use the Clarivate Analytics of the Web of Science Journal Citation Reports 2020, consistently.

Reply to Comments:

The impact factor values have been modified in the manuscript and appendix as suggested and are highlighted in yellow.

Line/Page number:

Line 154-156 (Table 1 in manuscript)

---

## [Editor Report · Decision Letter 3]

14 Dec 2021

Authors submission guidelines, a survey of Pediatric Dentistry journals regarding ethical issues.

PONE-D-21-18572R3

Dear Dr. Walia,

We’re pleased to inform you that your manuscript has been judged scientifically suitable for publication and will be formally accepted for publication once it meets all outstanding technical requirements.

Kind regards,

Despina Koletsi, Dipl.D.S, MSc, Dr. med. dent, MSc, DLSHTM, PGCHEd

Academic Editor

PLOS ONE
---

## [Editor Report · Acceptance letter]

7 Jan 2022

PONE-D-21-18572R3 

Authors submission guidelines, a survey of Pediatric Dentistry journals regarding ethical issues. 

Dear Dr. Walia:

I'm pleased to inform you that your manuscript has been deemed suitable for publication in PLOS ONE. Congratulations! Your manuscript is now with our production department. 

Kind regards, 

on behalf of

Dr. Despina Koletsi 

Academic Editor

PLOS ONE